# Multidisciplinary Management of Craniopharyngiomas in Children: A Single Center Experience

**DOI:** 10.3390/diagnostics12112745

**Published:** 2022-11-09

**Authors:** Giada Del Baldo, Sabina Vennarini, Antonella Cacchione, Dante Amelio, Maria Antonietta De Ioris, Francesco Fabozzi, Giovanna Stefania Colafati, Angela Mastronuzzi, Andrea Carai

**Affiliations:** 1Department of Onco-Hematology, Cell and Gene Therapy, Bambino Gesù Children’s Hospital, Scientific Institute for Reasearch, Hospitalization and Healthcare (IRCCS), 00165 Rome, Italy; 2Pediatric Radiotherapy Unit, Fondazione IRCCS Istituto Nazionale dei Tumori, 20133 Milan, Italy; 3Proton Therapy Center, Azienda Provinciale per i Servizi Sanitari (APSS), 38123 Trento, Italy; 4Department of Diagnostic Imaging Oncological Neuroradiology Unit, Bambino Gesù Children’s Hospital, IRCCS, 00165 Rome, Italy; 5Neurosurgery Unit, Department of Neurosciences, Bambino Gesù Children’s Hospital, IRCCS, 00165 Rome, Italy

**Keywords:** craniopharyngioma, proton therapy, surgery, quality of life, children

## Abstract

Background: Craniopharyngioma (CP) is a rare brain tumor involving the sellar region. The best management is still debated. Gross total resection (GTR) is considered the best option to improve recurrence-free survival, but considerable long-term sequelae with a significant impact on quality of life have been reported. Subtotal resection followed by radiotherapy achieves similar disease control compared to GTR with less complications. Methods: We retrospectively reviewed 10 pediatric patients affected by CP treated with partial resection and subsequent proton therapy (PBT). We reviewed visual, endocrinological, and neuropsychological data at baseline, after surgery, and after radiation for all patients. Results: At the time of diagnosis, visual impairment was detected in 70% of patients and endocrinological abnormalities in 50%. All patients were subject to one or more surgical procedures. Surgery had no impact on visual status; however, it caused a worsening of endocrine function in half of patients. After surgery, all patients underwent PBT, achieving a partial response in 7 out of 10 patients (70%), while stable disease was observed in the other three patients (30%) at a median follow-up of 78 months from the end of PBT. Both visual and endocrine deficits were stable after PBT, with neurocognitive performance scores unchanged from baseline. Conclusions: A conservative surgical approach followed by PBT represents a safe and effective strategy to manage CP and limit long-term sequelae.

## 1. Introduction

Craniopharyngioma (CP) accounts for about 1–3% of all central nervous system (CNS) tumors and is mostly observed in pediatric patients between 5 and 10 years [1,2,3], without gender predilection. Despite the indolent behavior of these tumors, patients affected by CP experience neurocognitive impairment and hypothalamic and endocrine dysfunction, significantly impacting quality of life (QoL). Gross total resection (GTR) has been reported to contribute to longer progression-free survival (PFS), but it is burdened with a significantly increased risk of morbidity. On the other hand, the risk of progression after incomplete resection ranges from 50 to 90% [4].

Recent data suggest that subtotal resection (STR) followed by radiation leads to similar disease control and overall survival (OS) as GTR with fewer complications [5,6,7]. However, concerns regarding radiation-induced toxicities such as vascular damage, cognitive deficiencies, and secondary malignancies are still present. Technological refinements in radiation planning and delivery have improved the conformality of radiation doses to target volumes, reducing doses in nearby normal tissues. Proton beam therapy (PBT) is emerging in the treatment of pediatric CP with encouraging results in terms of sparing critical organs [8,9,10]. However, the best timing to deliver radiation is still debated.

The purpose of our study was to retrospectively describe an institutional cohort of children with CP who were treated with partial resection and subsequent PBT and provide additional information regarding endocrine sequalae and visual and cognitive impairment.

## 2. Materials and Methods

We retrospectively reviewed data on consecutive patients with histology-proven CP diagnosed between 2015 and 2021 at Bambino Gesù Children’s Hospital in Rome. We chose 2015 as the study’s cut off because that is the year PBT was available for use in Italy.

The inclusion criteria for this study were the following: age range 0–18 years, surgically treated CP with evidence of progressive or residual disease after surgery, histological diagnosis of CP, and radiant treatment by protons. The medical records of patients included: demographic data; surgical details; treatment for hydrocephalus if present; radiation therapy details; visual and endocrinological assessment at diagnosis, after surgery, and 6 months after PBT; neuropsychological assessment at diagnosis, after surgery, and 12 months after PBT; and data outcome and last follow-up.

The study was approved by the Institutional Review Board (IRB) of Bambino Gesù Children’s Hospital and was conducted in accordance with the Helsinki Declaration. Written informed consent was obtained from all the patients or legal guardians.

### 2.1. Radiological Assessment

Brain magnetic resonance imaging (MRI) (Siemens Magnetom Skyra, Erlangen, Germany) at 3 Tesla was carried out using a standardized pediatric protocol including axial and coronal T2-weighted sequences, axial FLAIR, diffusion-weighted imaging (DWI), susceptibility-weighted imaging (SWI), and midline sagittal volumetric T2-weighted and pre- and post-contrast volumetric T1-weighted sequences. Magnetic resonance angiography was used to provide information on the relationship between the tumor and critical vascular structures.

Radiological disease progression was categorized as the growth of the solid and/or cystic components of known postoperative disease residuals. Radiological relapse was defined as evidence of disease following previous MRI evidence of a completely resected lesion.

Radiological response was defined as follows: partial response (PR) if a significant reduction in tumor size was detected at the last MRI follow-up and stable disease (SD) when disease remained substantially unchanged.

### 2.2. Surgery

All procedures were performed by a dedicated pediatric neurosurgical team. Indication for surgery was discussed during a multidisciplinary tumor board in all cases. The general aim of surgery was maximal safe surgical resection at the time of diagnosis. Of note preoperative imaging was always used to raise awareness about potential unresectable tumor areas; nonetheless, the final judgment was always performed intraoperatively after the direct microsurgical inspection of the tumor–brain interface.

Whenever the patient presented with mass effect symptoms, including hydrocephalus, from a cystic tumor component, an endoscopic transventricular intracystic catheter placement was performed to allow for cyst drainage. This maneuver was always followed by a resection procedure. The most frequent approach was pterional, including orbital roof displacement and/or dissection through the fronto-basal interhemispheric corridor in selected cases. Patients presenting favorable anatomical features underwent resection via an endoscopic transnasosphenoidal approach. Whenever a dorsal residual tumor was considered resectable, a transcallosal or trans-cortical transventricular approach was performed in a second-stage surgery through the transforaminal microsurgical corridor. 

Microsurgical procedures were performed with navigation assistance (Medtronic S7), intraoperative neurophysiologic monitoring (Cadwell), and intraoperative ultrasound imaging (BK medical 5000).

Concomitant hydrocephalus was treated whenever necessary by ventriculoperitoneal shunting.

The extent of resection and residual type were defined by postoperative MRI. 

### 2.3. Proton Beam Therapy

PBT was proposed to all school-aged and older patients with evidence of residual disease after maximal safe resection. PBT was delayed in children younger than 6 years who underwent a subtotal resection until evidence of radiological disease progression. In all cases, further surgical excision was evaluated before PBT and excluded because of unacceptable surgical risk. The choice of whether or not to perform PBT on a patient was dictated by a multidisciplinary discussion between several specialists (Figure 1) considering patient age, status of residual disease (solid and/or cystic) and its potential cytoreduction with a non-worsening surgery, disease progression, and pre-existing deficits pre and post-surgery.

All patients received 54 gray (Gy) equivalent in 30 fractions. The relative biological effectiveness (RBE) factor for protons of 1.1 (relative to 60Co) was employed and proton doses were expressed in terms of gray equivalent (GyRBE = proton Gy x 1.1) [11].

Target volumes were defined as follows: the contrast-enhancing solid lesion depicted on T1-weighted MRI was defined as the gross target volume (GTV) along with all cystic components of the tumor, visible in T2-weighted MRI. According to the currently published indications [12], clinical target volume (CTV) was generated by adding the surgical bed to GTV with a 3–5 mm uniform margin manually corrected in the proximity of anatomical barriers. The planning target volume (PTV) consisted of CTV, adding a safety margin of 3 mm to account for possible treatment uncertainties.

All patients were treated with active beam scanning PBT using 3 beams. In the case of cystic residual components at the time of irradiation, patients received weekly focused T2-weighted MRI evaluations to detect treatment-related size changes.

Patients were seen weekly during radiotherapy and every 3–6 months thereafter.

The acute and late effects of PBT were assessed according to the National Cancer Institute Common Toxicity Criteria (Version 5.0) [13].

### 2.4. Ophthalmological Assessment

Visual performance data were collected at the time of diagnosis, after surgery, and every six months after PBT. Assessments included visual acuity (VA), optical coherence tomography (OCT), and visual field (VF) testing.

The type of VA test was chosen based on the patient’s age and ability to cooperate [14,15]. The Teller Acuity Cards test of grating acuity (Stereo Optical, Chicago, Illinois, USA) was used for children aged 2–3 years old. E-charts and Snellen or numeral charts were chosen for patients 3–5 years old and >6 years old, respectively

OCT is a non-invasive imaging test for the precise measurement of retinal nerve fiber layer thickness (RNFL). The cut-off RNFL thickness reduction was considered to be predictive of visual loss if it was greater than 10% from baseline in one or more quadrants or global average [16]. OCT was performed in general anesthesia for non-collaborative patients.

VF was tested according to age-adapted methods performed by a pediatric ophthalmologist in cooperative patients (behavioral visual field screening test, Humphrey visual field analyzer, semi-automatic static Peritest, or Goldmann kinetic perimetry) [17,18,19,20].

### 2.5. Endocrinological Assessment 

Endocrinological assessment data were collected at the time of diagnosis, after surgery, and every six months after PBT.

Clinical and auxological data, including body mass index (BMI), were retrospectively collected. The BMI z-score was calculated by adjusting for age and gender in patients >2 and <20 years [21]. As previously reported by the Centers for Disease Control, obesity was defined as a BMI z-score  >1.64 (95th centile) and overweight as a BMI z-score >1.04 (85th centile).

The assessment of pituitary function included an evaluation of gonadotropin deficiency, thyrotropin deficiency, corticotropin deficiency, growth hormone deficiency (GHD), and diabetes insipidus.

Biochemical and hormonal data, including IGF-I, IGFBP-3, FT4, TSH, FSH, LH, testosterone, 17-beta-estradiol, ACTH, cortisol, glucose, serum and urinary electrolyte levels and plasma and urine osmolarity, were retrospectively collected.

The endocrine evaluation included provocative tests to investigate hypothalamic-pituitary function as previously described [22].

Data on hormonal replacement treatment were recorded for all patients when performed.

### 2.6. Neuropsychological Evaluation

Neurocognitive performance was evaluated at the time of diagnosis, post-surgery, and 12 months after PBT. Measures at baseline and follow-up included an age-appropriate Wechsler scale of intelligence (WISC IV version) in children between 6 and 16 years old and Wechsler preschool and primary scale of intelligence (WPPSI IV version) in patients under 6 years of age [23].

The Wechsler scales provide an estimate of global intellectual ability (full-scale IQ) and four composites: verbal comprehension index (verbal skills); perceptual reasoning index (nonverbal reasoning); working memory index (working memory, short-term memory, sustained attention, and auditory processing); and processing speed index (visual-motor coordination, attention, concentration, and the speed of mental processing). WPPSI provides subtest and composite scores that represent intellectual functioning in verbal and performance cognitive domains as well as a composite score that represents full-scale IQ [24]. 

### 2.7. Statistical Analysis

Clinical characteristics were summarized as numbers and percentages for qualitative data and as medians with ranges between the maximum and minimum observation for continuous variables.

Follow-up was calculated from the date of the end of the PBT to the date of last follow-up. Overall survival (OS) was measured from diagnosis to the date of last follow-up, and the distribution was estimated using the Kaplan–Meier method.

## 3. Results

A total of 19 patients with histologically confirmed CP were diagnosed and underwent 26 surgical procedures between 2015 and 2021 at our center. Nine patients were excluded from this study: seven children had undergone GTR and two patients with STR were treated with photons instead of PBT due to family refusal.

Finally, ten patients presented with the inclusion criteria and were included in this study. There were four female and six male patients, with a median age at diagnosis of 8.7 years (range, 3.31–13.33).

Patient and clinical characteristics are described in Table 1.

At the time of diagnosis, visual impairment was detected in 7 of 10 patients. The visual defect was unilateral in three cases and bilateral in four. One patient presented with GHD and four patients with panhypopituitarism at diagnosis. Obesity was present in three patients and one patient was overweight at the time of diagnosis. All patients presented an average IQ score, except for one who fell into the borderline class.

Two patients presented with hydrocephalus at diagnosis and a ventricular peritoneal shunt was implanted. The first surgical treatment at diagnosis reached partial removal in 7 of 10 patients, while cyst drainage was performed in 3 of 10 patients. In six patients, multiple surgeries were necessary due to residual disease progression, including resection and/or cystic drainage.

After surgery, five patients presented complete hypothalamic-pituitary hormonal deficit, whereas the patient with isolated GHD developed panhypopituitarism at one year from diagnosis, before PBT. Obesity occurred in one case after surgery and four patients were overweight. Vision status remained stable for all patients and new deficits were not detected at any time after surgery. 

After the first or additional surgeries, all patients underwent to PBT (54 Gy RBE). The reason for treatment was the progression of disease in 7 of 10 cases, whereas PBT was performed subsequently after surgery in three patients with residual disease.

The median time from diagnosis to PBT start was 24 months (range, 4.7–77.8). The median age at PBT was 10.71 years (range 6.1–16.6). The median follow-up was 28 months (range, 13.5–77.8) from the end of PBT.

All patients were alive at the last follow-up, with a median survival of 78.9 months (range 20.8–108.8).

At the first MRI re-evaluation after PBT, a PR was described in 7 out of 10 patients (70%) and an SD was detected in 3 out of 10 patients (30%). PR was maintained for a median time of 26.7 months (range, 13.5–75.6) and SD for 41.3 months (range, 25.6–77.9).

No acute side effects were registered during PBT, and all patients completed the treatment without interruptions. Moreover, all MRI monitoring of the cyst component during proton treatment did not reveal any significant dimensional changes; thus, we did not have to change our CTV, and target coverage was always maintained. Both visual and endocrine defects were stable before and after PBT. No vascular impairment was found. Moreover, no deflection in neurocognitive performance score was detected one year after the end of PBT (Figure 2).

Detailed single-patient characteristics are described in Table 2.

Among the nine patients who were excluded from this study but treated in the same observational period at our institution, seven underwent only GTR, whereas two patients with STR were treated with photons instead of PBT.

At baseline, endocrine function was normal in three patients treated with GTR, while one already had panhypopituitarism. All these patients developed panhypopituitarism after surgery. For the two patients treated with STR and photons, one already had panhypopituitarism at diagnosis and one developed it after surgery.

Visual function at baseline was impaired in all but one of the seven patients treated with GTR. Visual function improved after GTR in only one patient, while one was affected by complete blindness. The two patients treated with STR and photon therapy presented visual impairment in both eyes at diagnosis and the deficits were stable after surgery and radiation. 

Obesity was never detected at diagnosis, but it occurred in four out of nine patients (45%) after GTR.

Lastly, neurological deficits occurred in two patients, one treated with GTR and one with STR. 

Unfortunately, complete data regarding neurocognitive performance were collected only for few patients. 

## 4. Discussion

CP is the most common type of tumor in the sellar region; nevertheless, the optimal therapeutic strategy for CP remains controversial. Although the mortality rate of CP is very low, treatment-related morbidity can have a devastating impact on physical, social, emotional, and cognitive functions.

Surgical resection represents a cornerstone in the treatment of CP. Safe GTR remains the gold standard when feasible [25] and is associated with a lower risk of recurrence, described in less than 50% of patients [26]; however, it is burdened by high rates of optic and endocrinological impairment, lower QoL score [27], and a loss of full-scale intelligent quotient points compared to incomplete resection [7]. STR is associated with reduced postoperative complications, but an increased rate of recurrence has been described; progression at 5-years follow-up after incomplete resection occurs in 71–90% of patients [4].

The intricate anatomy of the sellar and parasellar region and the tendency of CP to extend to adjacent brain compartments, such as the ventricular system and the anterior, middle, and posterior cranial fossae, require excellent surgical skills and versatility to choose the best approach to the lesion [28].

The pendulum swings between striving to reach a complete resection with severe long-term and permanent adverse effects [29,30,31] and aiming for STR with the preservation of the hypothalamus but with the need for additional surgeries and/or radiation treatment to obtain disease control [30].

In our study, we reported ten CP pediatric patients treated with PBT after partial removal or at progression. PBT presents limited toxicities and mainly stable visual, endocrinological, and cognitive impairment after treatment.

In our experience, multiple procedures were performed on most of the patients to delay the need for radiation, based on its well-known detrimental effects on the developing nervous system. Whatever the approach, the conundrum of CP resection remains the possibility of safely dissecting the tumor margin from the surface of the hypothalamus. Despite attempts to discriminate between the compression and invasion of the hypothalamus, the intraoperative impression of the operating surgeon still remains the most relevant factor in determining the feasibility of a complete resection. In our series, we favored the transcranial pterional approach. The inclusion of the anterior orbital roof in the craniotomy allows for a wider exposure of the tumor dome and can be combined with additional surgical corridors to tailor the exposure of the tumor. We considered the trans-sphenoidal route only in select cases. Major determinants favoring the transnasal approach included a sufficient intercarotid working channel and the absence of significant extension in the suprasellar compartment. Unless a resection of the tumor was planned, we approached cystic lesions by the stereotactic endoscope-assisted positioning of intracystic catheters in an effort to minimize the manipulation of nervous tissue.

Despite surgical treatment and its related potentially severe complications, the risk of progressive disease is not negligible, even in the case of complete removal [25]. The addition of adjuvant radiotherapy reduces this recurrence rate. Multiple studies have demonstrated that the addition of RT to STR reduced recurrence rates by about 20% [4,32], which is similar to the recurrence rates achieved after GTR in multiple series [33,34,35,36]. These data suggest the superiority of GTR and STR followed by RT over STR monotherapy with regard to recurrence rates. However, both aggressive surgery and RT carry a high risk of long-term sequelae. The therapeutic consequences of radiation-induced toxicities include visual and endocrinological injuries, vascular changes, cognitive deficiencies, and secondary malignancies [37]. For this reason, technologic improvements in radiation delivery have attempted to reduce the doses delivered to local normal tissues with the aim of decreasing secondary sequelae [37]. Unfortunately, the absence of prospective randomized trials due to a small number of patients and the difficulty of a suitable enrolment based on patient selection characteristics makes it difficult to carry out an objective therapeutic comparison of different approaches. Regrettably, the choice of radiation type is often at the discretion of the experience of the single center and is influenced by several factors, such as the age of the patient, the status of residual disease, and pre-existing deficits pre- and post-surgery [7,38,39,40,41]. Numerous external beam radiotherapy techniques have been employed with conventional fractionation and stereotactic hypofractionation radiotherapy to complement subtotal surgery, all of which have demonstrated excellent results in terms of local disease control and a reduction in CTV expansion margins with the sparing of local tissue to the target of irradiation [42,43]. The technological evolution of radiotherapy has increasingly led to the use of accurate and precise techniques with reduced doses delivered to healthy tissue, such as PBT, for which CP is the elective indication [44]. Boehling et al. conducted comparative photon-proton dosimetric studies with advanced intensity-modulated techniques (IMPT), sparing both irradiated vascular and hippocampal systems in favor of proton therapy [9]. The recognized dosimetric approach for normal fractionated therapies provides a total dose of 50.4–54 Gy fractioned in 1.8 Gy per session. This dose prescription perfectly respects the threshold radiobiological tolerance of the major organs at risk adjacent to the target volume, such as the chiasm and brainstem [45]. Moreover, a preliminary study by Luu et al. reported a local disease control rate with PBT of about 90% [46], nearly equivalent to the historical small photon cohorts published in [36], with a reduced risk of sequelae compared to radiotherapy and likely the same rate of effectiveness.

In our experience, PBT was performed in all patients after one or more surgeries and was always well-tolerated. No patients presented progressive disease after PBT, confirming its effectiveness with a high rate of local disease control. In fact, we described a PR and SD rate of 70% and 30%, respectively, at a median follow-up of 28 months after the end of PBT.

Moreover, it is important to highlight the relevance of monitoring the cystic component, which can increase during irradiation and even after proton therapy has ended. Recognizing transient cyst growth is critical for sparing unnecessary patient intervention. In our experience, the MRI monitoring of the cyst component during proton treatment did not reveal any significant dimensional changes; thus, we did not have to change our CTV, and target coverage was always maintained [47].

In addition to patient age, disease volume, and residual type, one of the criteria of choice for surgical strategy and radiation therapy was also related to the patient’s clinical status at the time of treatment. It is well-known that patients with CP may already present with endocrinological and/or visual defects at diagnosis. Endocrinopathy is the most frequent symptom described at the onset of the disease (90% of cases) [48]. GHD is the most common endocrine disorder (70%), followed by central hypogonadism (51.7%). Although rare, thyroid and corticosteroid deficits may also coexist (25%). Central diabetes insipidus is reported in about 28% of patients [48,49,50]. Obesity and eating disorders are described in about 50% of children with CP [31,51,52,53,54,55,56,57], and the degree of obesity is strongly related to the extension of hypothalamic involvement [31,58,59]. In our series, endocrinological dysfunction was detected in 50% of patients at diagnosis, including isolated GHD in one case and panhypopituitarism in four cases. Obesity was present in 30% of patients and only one patient was overweight.

In more than 50% of patients affected by CP, visual acuity and fields can deteriorate [60], causing difficulties in scholarly activities, daily life, and self-perception [61]. We found visual impairment in 70% of our patients at the time of diagnosis.

The risk of presenting hypopituitarism, obesity, and visual impairment increases with surgery [62]. Bakhsheshian et al. reported a case description of 1961 pediatric patients affected by CP who underwent a transcranial or a trans-sphenoidal craniotomy. The most common post-surgical complication described was diabetes insipidus (64%), with no independent factors associated [63]; however, other hypothalamic pituitary endocrinological dysfunctions, including obesity, were reported. Other studies reported irreversible central diabetes insipidus in 80–93% of all complete resections and GHD in 75% of cases [64]. Approximately 80–90% of patients who undergo surgery develop panhypopituitarism, and deficits in four or more hormones have been reported in a large portion of patients. The frequency of new pituitary hormone deficiency seems to be higher in patients who underwent transcranial surgery rather than the trans-sphenoidal approach [48,65,66]. Moreover, obesity was described in about 65% of long-term CP patients who underwent surgery.

In relation to this study, none of our patients had a significant worsening of their visual performance after surgery. This notable result might depend on the function preservation-oriented surgical style of our group. This is of paramount importance, since these children have a lifelong risk of becoming visually impaired. Regarding endocrinopathy, half of our patients already had an endocrinological defect before surgery, and panhypopituitarism was observed in the remaining patients. According to data in the literature, hormonal defects are the most frequent post-surgical deficit. In our experience, post-surgery panhypopituitarism was not related to multiple surgeries; moreover, hormonal replacement therapy was always tolerated and does not represent a limitation to the maximal safe resection of the tumor.

Despite these findings, identifying long-term complications related to surgery is not easy and their description varies from study to study. As for surgery, it is very difficult to establish and distinguish long-term toxicities related to radiotherapy rather than to the disease itself. The most common effect related to radiotherapy is the worsening of endocrine dysfunction, which is observed in 77–95% of patients; induced panhypopituitarism is observed in 30–46% of patients [5,37,67,68]. Hypothalamic obesity after combined treatment is also uncertain but expected and is found in around 25–55% of patients [37,69,70,71]. Visual decline as a result of long-term toxicity is rather rare with modern radiotherapy techniques and in particular with proton therapy, as is the case for neurocognitive impairment and cerebrovascular changes. Mueller et al. [72] published data showing that the incidence of stroke and vascular abnormalities are doubled in childhood tumors. In other studies focused on CP patients, *moya-moya* vascular complication rates were found to be around 10% at long-term follow-up [37,73,74,75].

In our experience, no acute side effects were reported during radiation. At last follow-up, hormonal, visual, and neuropsychological status overlapped with pre-operative evaluation for all patients. Moreover, vascular complications were never reported.

Because the morbidities associated with aggressive surgery or RT can be particularly pronounced in pediatric patients, the most appropriate management of CP in this age group remains controversial. Despite the many efforts and the scientific research conducted to date, open problems still remain, especially in the case of post-radiation progression/recurrence or in the youngest patients, in which it would be best to avoid or postpone radiation treatment because of the known related complications. Surgical management in children remains controversial and must be carefully planned. One of the major surgical topics discussed concerns the pre-surgical stratification of patients to determine the best surgical planning to reduce morbidity and spare the hypothalamus [25]. The consideration of a combined approach with radiation in the case of partial removal and the right timing to start irradiation is still debated, mainly in the youngest patients. Specifically, additional therapeutic approaches, such as intracystic therapy and target therapies, have been developed in recent years; however, the available studies are small and with limited data to support their use [25]. Therefore, it becomes essential to consider alternative treatment and extend molecular studies to detect a potential target that may be useful in the future. The main signal pathways involved in CP development are summarized in Figure 3.

Although our case series was very small, our work demonstrates that a conservative approach followed by proton radiation is a valuable choice for limiting postoperative complications. PBT is safe and confers optimal disease control. To protect the patient’s QoL and avoid long-term sequelae, hypothalamus-sparing treatment approaches are recommended and radiation should be considered immediately after surgery because tumors tend to fail quickly without further treatment, despite the fact that in clinical practice, the timing of postoperative residual tumor irradiation is still unclear. However, long-term endocrine and visual follow-up is necessary, and neuropsychological testing may identify patients at risk for treatment-related cognitive and adaptive functioning changes.

Based on our data, we recommend performing the safest surgery; if a GTR is not feasible, adjuvant treatment with radiation should be considered, using the best technologies available to date (such as PBT) acting on the residual disease to consolidate the result obtained with surgery. STR combined with proton therapy on the residual disease seems a wise therapeutic strategy for the treatment of CP in terms of morbidity and outcome, providing a fair balance between the risks of sequelae and recurrence. Alternative therapies are emerging, but at this moment are ineffective in obtaining good disease control and an acceptable safety profile.

## 5. Conclusions

Our experience advises that risk-adapted surgical strategies at diagnosis should aim for a maximal degree of resection, respecting the integrity of optical and hypothalamic structures to prevent severe sequelae and therein minimize consequences that could negatively impact the patient’s QoL. STR followed by PBT results in excellent disease control for pediatric CP, with a low risk of acute toxicity. Notably, PBT does not seem to worsen pre-existing complications, namely vision, the neurocognitive domain, and endocrine function, resulting in a safe and effective option for CP management.

Our preliminary experiences with PBT applied to operated CP are very promising compared to conventional irradiation, especially for localized tumors adjacent to the optic nerve/chiasm, pituitary gland, or hypothalamus. Long-term follow-ups are needed to confirm this data.

## Figures and Tables

**Figure 1 diagnostics-12-02745-f001:**
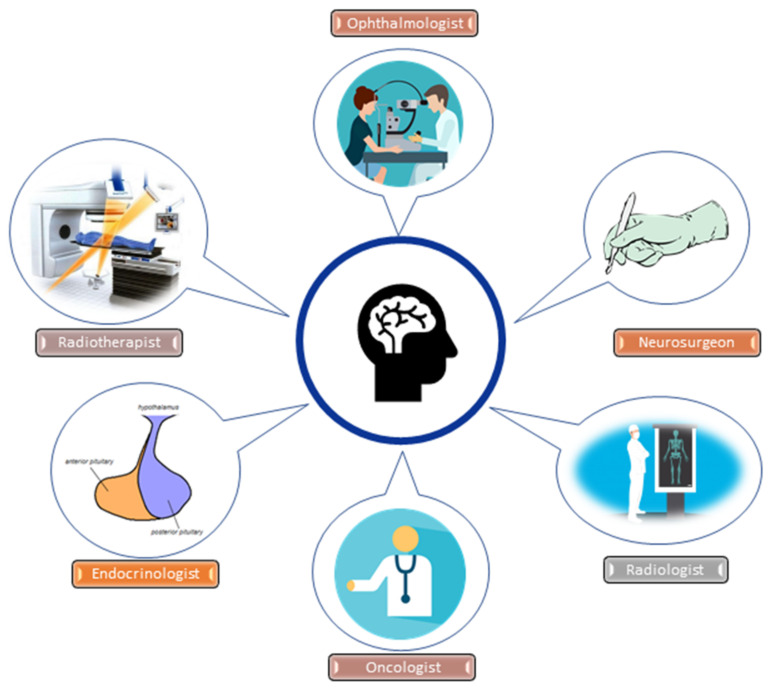
Multidisciplinary assessment in pediatric craniopharyngioma management.

**Figure 2 diagnostics-12-02745-f002:**
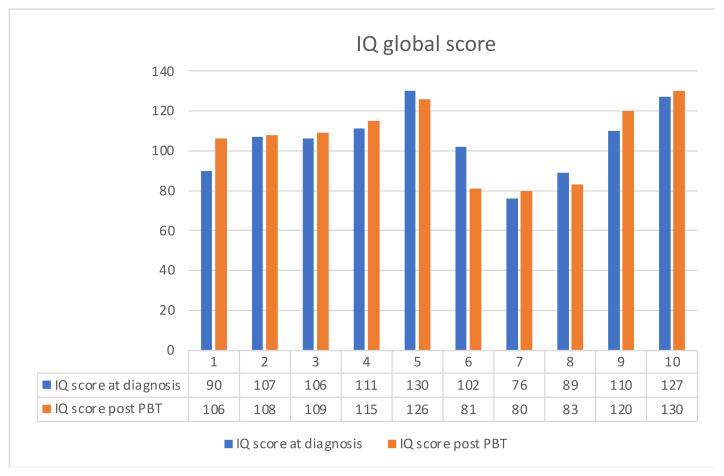
Neurocognitive performance score at diagnosis and after PBT.

**Figure 3 diagnostics-12-02745-f003:**
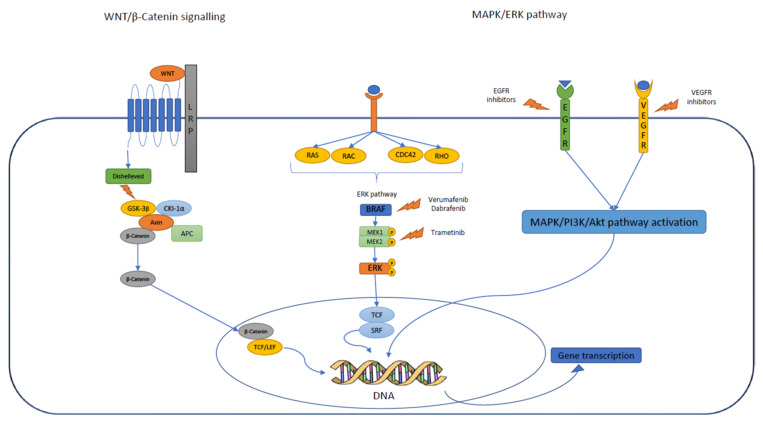
Molecular pathways involved in CP development and potential target therapies.

**Table 1 diagnostics-12-02745-t001:** Population characteristics, treatment, and follow-up.

Characteristics	*n* (%)
Age at diagnosis	Median 8.7 years (range, 3.3–13.3)
Gender	Male 6 (60%); female 4 (40%)
Clinical characteristics at diagnosis:	
Endocrine impairment	GHD: 1 (10%); panhypopituitarism: 4 (40%)
Obesity	3 (30%)
Visual impairment	7 (70%)
Surgical details:	
Shunt implant for hydrocephalus	2 (20%)
Single surgical approach	4 (40%)
Multiple surgeries	6 (60%)
PBT details:	
Age at PBT	Median 10.7 years (range, 6.1–16.6)
Relapse/progression after surgery	7 (70%)
Residual disease after surgery	3 (30%)
Residual type before PBT	Solid: 2 (20%)
	Cystic-solid: 8 (80%)
Time from diagnosis to PBT	Median 24.17 months (range, 4.7–77.8)
Follow-up data:	
Status after PBT	SD: 3 (30%); PR: 7 (70%)
Median duration PR	26.7 months (range, 13.5–75.6)
Median duration SD	41.3 months (range, 25.6–77.9)
Survival	78.9 months (range, 20.8–108.8)

**Table 2 diagnostics-12-02745-t002:** Patient characteristics at diagnosis, post-surgery, and post-PBT.

Patient	Age at Diagnosis (Years)	Sex	Visual Impairment at Diagnosis	Endocrine Dysfunction at Diagnosis	HC	More than One Surgery before PBT	Status before PBT	Visual Impairment after Surgery, before PBT	Endocrine Dysfunction after Surgery, before PBT	Age at PBT (Years)	Status after PBT (Duration in Months)	New Visual Impairment after PBT	New Endocrine Dysfunction after PBT	OS
1	6.2	M	YES (in one eye)	PH	NO	NO	PD	YES (in one eye)	PH	6.8	SD (77.9)	NO	NO	86.9
2	12.7	M	NO	PH	NO	NO	PD	NO	PH	14.2	PR (75.5)	NO	NO	95.2
3	9.3	F	NO	GHD	YES	YES	PD	NO	PH	11.1	SD (25.6)	NO	NO	48.2
4	8.1	F	YES (in one eye)	PH	NO	NO	PD	YES (in one eye)	PH	10.3	PR (13.5)	NO	NO	42.2
5	2.6	M	YES (in both eyes)	PH	NO	YES	PD	YES (in both eyes)	PH	8.2	PR (41.8)	NO	NO	108.9
6	13.3	F	YES (in both eyes)	NO	YES	YES	RD	YES (in both eyes)	PH	15.7	SD (40.9)	NO	NO	71
7	11.1	M	YES (in one eye)	NO	NO	YES	RD	YES (in one eye)	PH	16.6	PR (29.2)	NO	NO	97.2
8	4.4	F	YES (in both eyes)	NO	YES	YES	PD	YES (in both eyes)	PH	6.1	PR (26.7)	NO	NO	43.9
9	3.3	M	YES (in both eyes)	NO	NO	YES	RD	YES (in both eyes)	PH	9.8	PR (26.8)	NO	NO	105.9
10	11.2	M	NO	NO	NO	NO	PD	NO	PH	11.6	PR (14.7)	NO	NO	20.8

Legend: F—female; GHD—growth hormone deficiency; HC—hydrocephalus; M—male; OS—overall survival; PBT—proton beam therapy; PD—progressive disease; PH—panhypopituitarism; PR—partial remission; RD—residual disease; SD—stable disease.

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
