# Peer review of "Multidisciplinary Management of Craniopharyngiomas in Children: A Single Center Experience"

_diagnostics, 2022, doi:10.3390/diagnostics12112745_

Round 1

Reviewer 1 Report

1.      Baldo et al report their institutional data on combination subtotal resection and adjuvant protein beam radiation for the treatment of pediatric craniopharyngioma. They case series reports results for a total of 10 pediatric patients with approximately 7 year follow up highlighting their overall survival, progression free survival, endocrinological data, visual, and neuropsychiatric data.

2.      Strengths: The study is unique in that it is one of the few if only studies that show neurocognitive data in this patient population. Weaknesses: Limited sample size, no comparison to GTR patients or patients with STR treated photo modalities, lack of clarity of progression of visual or endocrine symptoms following PBT

3.      Recommendations

a.      Requires major changes

                                                    i.     Should consider providing comparison to GTR and STR with conventional radiation therapy populations to contextualize data as safety and efficacy is contextual to this data

                                                   ii.     Should consider providing more granular neuropscyh data

b.      Minor changes

                                                    i.     Typo for “QI” instead of “IQ”

                                                   ii.     Adjust visual impairment after PBT to NEW visual impairment after PBT (should all be no)

                                                  iii.     Same adjustment as above for endocrine dysfunction column

Author Response

  1. Del Baldo et al report their institutional data on combination subtotal resection and adjuvant protein beam radiation for the treatment of pediatric craniopharyngioma. They case series reports results for a total of 10 pediatric patients with approximately 7 year follow up highlighting their overall survival, progression free survival, endocrinological data, visual, and neuropsychiatric data. Strengths: The study is unique in that it is one of the few if only studies that show neurocognitive data in this patient population. Weaknesses: Limited sample size, no comparison to GTR patients or patients with STR treated photo modalities, lack of clarity of progression of visual or endocrine symptoms following PBT.  Recommendations

a. Requires major changes

i. Should consider providing comparison to GTR and STR with conventional radiation therapy populations to contextualize data as safety and efficacy is contextual to this data

A paragraph describing the data in patients undergoing GTR and STR followed by conventional radiotherapy was included in the Results section.

ii. Should consider providing more granular neuropscyh data

 Figure 2 has been modified by adding the precise numerical data of IQ.

b. Minor changes

i. Typo for “QI” instead of “IQ”

The typo was corrected.

ii. Adjust visual impairment after PBT to NEW visual impairment after PBT (should all be no)

Table 2 was modified as suggested.

iii.  Same adjustment as above for endocrine dysfunction column

Table 2 was modified as suggested.

Reviewer 2 Report

The conclusion is very feeble. I expect the writers to make the conclusion very comprehensive. Include the patient's symptoms and all relevant data in conclusion. Showcase your brilliance in the conclusion. The conclusion is weak. The writers have the opportunity to make it 2 paragraph conclusion. Make it 1st class and comprehensive.

Author Response

  1. The conclusion is very feeble. I expect the writers to make the conclusion very comprehensive. Include the patient's symptoms and all relevant data in conclusion. Showcase your brilliance in the conclusion. The conclusion is weak. The writers have the opportunity to make it 2 paragraph conclusion. Make it 1st class and comprehensive.

The conclusion was further elaborated to make the results more appealing

Reviewer 3 Report

  • Cases collected should include pathology with disease but without surgery as a control for comparison.

Author Response

  1. Cases collected should include pathology with disease but without surgery as a control for comparison.

Unfortunately, no patients were treated without surgery at our Institution in the same observational period.

Round 2

Reviewer 3 Report

  This article analyzes the surgical treatment of craniopharyngioma and draws the useful conclusion that STR combined with proton therapy for residual lesions appears to be a sensible treatment strategy for CP. This paper showed that PBT was performed after surgeries and no patients presented progressive disease after PBT confirming effectiveness. This paper is useful for the treatment of this disease and can be published.